# Enhancing the Properties of Polyvinyl Alcohol Films by Blending with Corn Stover-Derived Cellulose Nanocrystals and Beeswax

**DOI:** 10.3390/polym15214321

**Published:** 2023-11-04

**Authors:** Namhyeon Park, Mason A. Friest, Lingling Liu

**Affiliations:** 1Department of Agricultural and Biosystems Engineering, Iowa State University, Ames, IA 50010, USA; 2Department of Plant Pathology and Environmental Microbiology, Pennsylvania State University, University Park, PA 16802, USA; 3Department of Mechanical Engineering, Iowa State University, Ames, IA 50010, USA

**Keywords:** biodegradable, biopolymer, nanocomposites, polymer blend, water vapor transmission rate

## Abstract

Coating is a technique to surround a target substance with a thin layer to obtain desirable properties. Polyvinyl alcohols (PVAs) are biodegradable plastics and have shown good applicability as a coating or film material. Cellulose nanocrystals are a promising green nanomaterial that has been shown to enhance the properties of PVA after blending. However, these PVA/CNC films have concerns in a moist environment due to high hydrophilicity. To overcome this issue, the current study incorporated beeswax into PVA/CNC films and investigated the effect of CNC and beeswax on the properties of the coatings and films. Results showed that the addition of corn stover-derived CNCs to PVA films increased tensile strength (from 11 to 25 MPa) and Young’s modulus (from 32 to 173 MPa) and reduced water vapor transmission rate (from 25 to 20 g h^−1^ m^−2^). Beeswax added to PVA/CNC films further improved water vapor barrier properties (from 20 to 9 g h^−1^ m^−2^) and maintained Young’s modulus (from 173 to 160 MPa), though it caused a reduction in the tensile strength (from 25 to 11 MPa) of the films. This information can help to select materials for blending with PVAs by obtaining the desirable endmost properties depending on applications.

## 1. Introduction

Coating is to incorporate a thin layer of materials into a substrate by deposition in either the liquid phase (solution) or the solid phase (powder or nanoparticles) [1]. The coating technique can impart different attributes (e.g., mechanical integrity) to target substances to improve stability and resistance by acting as a physical barrier against environmental stresses [2]. Because of their benefits and applicability, coating is widely employed at various industries [3], including food and agricultural fields. 

General plastic film materials have good barrier and mechanical properties, but they do not automatically degrade in a short period of time, raising consumers’ concerns worldwide. Polyvinyl alcohol (PVA), on the other hand, is a water-soluble biodegradable plastic due to possible degradation by bacteria [4]. PVAs have high compatibility with other natural biopolymers for the production of coating films as they generate intermolecular hydrogen bonds and reduce the affinity to water molecules [4]. These natural biopolymers are of interest as coating or film materials because of their nontoxicity, relatively low price, and good quality [5,6], although their barrier properties can be lower than synthetic polymers. In this context, polymer blends can be an alternative approach to formulate desired properties according to their application. Each polymer has different properties with unique strengths and weaknesses, so utilization of a proper combination of these polymers can improve their application potential. 

Cellulose nanocrystals (CNCs) have been reported as one of the potential biopolymers for blending with PVAs, according to previous studies [7,8]. CNCs can be extracted from crystalline regions of cellulose [9], which consist of ordered (crystalline) regions along with some disordered (amorphous) regions. Several different processing methods, including mechanical, chemical, and enzymatic treatments, can produce CNCs with lightweight, nontoxicity, high mechanical strength, high surface area, biodegradability, and biocompatibility [10]. In addition, CNCs can be derived from agricultural wastes, making them sustainable materials [11]. These CNC compounds could impart higher stiffness in polymer matrices due to their large aspect ratio and ability to form interconnected network structures through hydrogen bonding [12].

The potential concern of coating films made of PVAs and CNCs was their highly hydrophilic properties, limiting their application in a moist environment [13], such as agricultural fields. This suggests the need for hydrophobicity in the final coating films to remedy their shortcomings because hydrophobic coating material lacks the affinity between the film layer and water [14]. The utilization of hydrophobic wax can mitigate the hydrophilicity of biopolymers and improve the barrier property against moisture (water vapor) [15,16]. Wax itself is a protective coating for some fruits in nature, among which beeswax and carnauba wax are the most studied [17]. Beeswax is a relatively abundant and edible natural product [15]. This product is inherently hydrophobic as it is a mixture of over 300 hydrocarbons, free fatty acids, fatty acid esters, fatty alcohols, diesters, and exogenous substances [18]. In addition, beeswax possesses strong water barrier properties [15] and can mitigate the hydrophilicity of final coatings. To our best knowledge, limited studies are available on the effect of the combination of CNC and beeswax on the properties of synthetic biopolymer films. 

This current work is to analyze the effect of corn stover-derived CNCs and beeswax on the final property of PVA films. CNCs derived from corn stover were characterized before its blending with PVA films. Properties of coating solutions consisting of PVA, CNCs, and beeswax were studied, including particle size, and viscosity. Properties of PVAs/CNCs/beeswax coating films were further characterized by UV-Vis, optical microscope, tensile tests, and water vapor barrier tests to evaluate their compatibility. 

## 2. Materials and Methods

### 2.1. Materials

Corn stover was obtained from the Iowa State University BioCentury Research Farm (Boone, IA, USA). Sulfuric acids (95–98%) (chemPUR, Karlsruhe, Germany), glacial acetic acids (Fisher scientific, Hampton, NH, USA) and sodium hydroxide (Macron Fine Chemicals, Center Valley, PA, USA), sodium chlorite (Beantown chemical, Hudson, NH, USA), sodium chloride (VWR International, Radnor, PA, USA), calcium chloride dihydrate (Sigma Aldrich, St. Louis, MO, USA), beeswax (Ward’s Science, Rochester, NY, USA), Glycerol (capitol scientific, Austin, TX, USA), polystyrene cuvettes (Fisher scientific, Hampton, NH, USA), PTFE Petri dishes (Fisher scientific, Hampton, NH, USA), PVA (Acros organics, Geel, Belgium), and dialysis tubings (Cellulose, 12–14 kDa MWCO) (Spectrum, Laguna Hills, CA, USA) were used. NPK (16-4-8) fertilizer was a gift from Timac Agro USA Inc (Reading, PA, USA). 

### 2.2. Preparation and Characterization of CNCs

#### 2.2.1. Preparation from Corn Stover

Dried and milled corn stover was treated with a NaOH solution (4%, *w/w*) for 4 h at 80 ± 0.2 °C in a shaking water bath at 120 rpm. After washing with DI water until the filtrate became clear, the alkaline treated corn stover was dried at 60 ± 0.2 °C for 24 h. Then, the dried samples (40 g) were bleached with a solution (24 mL of acetic acid and 30 g of NaClO_2_ in 746 mL of DI water) at 80 ± 0.2 °C for 2 h at 120 rpm. The bleached samples were washed repeatedly using DI water until the pH of DI water used became neutral. The whole steps to bleach and wash were repeated twice for the same samples, and the resulting bleached samples were dried at 60 ± 0.2 °C for 24 h. The dried and bleached samples (30 g) were milled and treated with 600 mL of 63% (*w/w*) sulfuric acid solution at 45 ± 0.2 °C for 45 min under stirring at 120 rpm. After the hydrolysis, the suspension was diluted 10-fold with cold water to stop the hydrolysis reaction and centrifuged at 15,821 × *g* for 20 min to remove the excess acid. The precipitate was then dialyzed with DI water until the pH of the dialysate was the same as that of DI water. Subsequently, the resulting CNC suspension was stored at 4 ± 1 °C. The average solid content (%, *w/w*) of the CNC suspension was measured based on the weight differences before and after drying at 105 ± 2 °C. The yields (step and overall yields) of the corn stover sample were calculated according to a previous method [19] and the equations are shown below.
(1)$$\%\;\mathrm{Step}\;{\mathrm{yield}}_{\;}=\frac{Dried\;sample\;weight\;after\;a\;certain\;treatment}{Dried\;sample\;weight\;before\;a\;certain\;treatment}\times 100$$

% Overall yield_After treatment *n*_ = Step yield_1_ (%) × Step yield_2_ (%) × … × Step yield*_n_* (%) (2)
where n = 1, 2, 3, and 4.

#### 2.2.2. Color Measurement

Different corn stover samples in each step of preparation was measured by a colorimeter (3nh NR-10QC, Shenzhen THREENH Technology Co., Ltd., Shenzhen, China)

#### 2.2.3. Transmission Electron Microscopy (TEM)

The morphology and size of CNC was measured using TEM according to a previous method [20]. In brief, the diluted CNC sample was observed using a 200 kV JEM-2100 scanning/transmission electron microscope (JEOL Co., Ltd., Akishima, Tokyo, Japan).

### 2.3. Preparation and Characterization of Coating Solutions 

#### 2.3.1. Preparation 

PVAs (3 g) were dissolved in 50 mL of DI water at 90 ± 1 °C for 1 h with magnetic stirring at 100 rpm. Then, glycerol (0.6 g) with different amounts of (3.75, 7.5, and 15 g) of CNC suspensions (2%, *w/w*) was added to the PVA solution at 90 ± 1 °C with constant stirring. DI water was then added until the final volume of each solution became 100 mL. After complete melting, 1.5 g of beeswax was placed into the solutions and mixed under vigorous stirring (200 rpm) at 90 ± 1 °C for 1 h. The resulting solutions in an ice bath were sonicated at 100% amplitude by using a 500 W ultrasonicator (Fisher Scientific, Hampton, NH, USA) for 5 min and degassed for 5 min by vacuum until no bubbles were observed. As a control, the solutions without CNCs and/or beeswax were prepared, and the same processing was employed. The composition of the final samples is indicated in Table 1. 

#### 2.3.2. Droplet Size Analysis 

The droplet sizes of prepared coating solutions were analyzed by dynamic light scattering using a Malvern Zetasizer Nano ZS instrument (Malvern Panalytical Co., Ltd., Worcestershire, UK). The measurements were performed at room temperature, and each droplet size distribution was obtained by averaging 3 different measurements.

#### 2.3.3. Viscosity Assay 

The viscosity of each coating solution was measured using an HR-2 Discovery rheometer (TA Instrument, New Castle, DE, USA). The measurements were conducted using a cup and bob geometry. The diameters of cup and bob geometry were 30.4 mm and 28 mm. After adding 25 mL of each sample, a flow ramp test at room temperature (23 °C) was performed. The viscosity was recorded in the shear rate ranging between 1 to 100 s^−1^, and the measurement time was 60 s.

### 2.4. Preparation and Characterization of Coating Films

#### 2.4.1. Preparation 

The freshly prepared emulsions (20 g) were cast on PTFE petri dishes and were dried in a fume hood at room temperature for 2 days. The films were stored in a desiccator for further characterizations.

#### 2.4.2. Thickness Measurement of Coating Films

After drying in a fume hood for 48 h, each coating film was peeled from the plates. The thickness of different films was measured using an electronic digital caliper (accuracy: ± 0.02 mm) (Neiko tools USA, Greenacres, FL, USA ).

#### 2.4.3. Optical Microscopy

The surfaces of coating films were analyzed using a 50W Halogen Trinocular Microscope (Amscope, Irvine, CA, USA).

#### 2.4.4. UV-Vis

Transmission of the films was taken at three different locations on the film equally spaced apart from the center using a SM1800 UV-VIS spectrophotometer (Azzota Scientific, Claymont, DE, USA) at wavelengths ranging from 200 to 800 nm.

#### 2.4.5. Tensile Testing

Tensile test measurements were taken according to ASTM D882 using an Instron 5944 (Instron, Norwood, MA, USA) with pneumatic grips and a 50 N load cell. Samples were cut into a dog-bone shape with a grip section dimension of 1 cm × 5 cm and a thin section (placed between the grips) of 0.6 cm × 2 cm. The film thickness was around 0.12 mm. Films were conditioned for 1 day at 50% relative humidity. The machine was set to a crosshead speed of 20 mm min^−1^, and the separation was zeroed before each run. Tests were conducted at room temperature. Averages were taken from 3 runs per batch, and values for tensile strength and Young’s Modulus were calculated from the Instron software (Bluehill 3) (Instron).

#### 2.4.6. Water Vaper Permeability of Coating Films 

The water vapor permeability of films was measured according to a previous method [13] with modifications. In order to control relative humidity (RH), NaCl saturated solutions (75% relative humidity at 22 °C) were placed in the bottom of the test chamber. To maintain a 0% RH in weighing bottle, 1.5 g of anhydrous CaCl_2_ was added. Each coating film tested was placed over the mouth of the weighing bottle and sealed with molten beeswax over the rim of the weighing bottle. Then, the weighing bottle was placed in the test chamber (75% RH and 22 °C) and was weighed after 24 h. The water vapor transmission rate (WVTR) was calculated according to the following equation.
(3)$$Water\;vapor\;transmission\;rate=\frac{\Delta w}{\Delta tA}$$
where Δ*w*/Δ*t* was rate of water gain (g h^−1^) and *A* was the exposed area of the film (m^−2^).

### 2.5. Statistical Analysis 

All experiments were performed with three replicates, and results were expressed as mean ± standard deviation. The results for color measurement, mechanical properties, and WVTR were analyzed by one-way ANOVA using PROC GLM after confirming normal distribution and homogeneity of variance (fixed factors: coating film formulations) followed by Tukey’s post hoc analysis (the significant difference was set at *p* < 0.05) (SAS version 9.4, SAS Institute Inc., Cary, NC, USA).

## 3. Results and Discussion

### 3.1. Yields and Color of CNCs Prepared from Corn Stover

Corn stover comprises cellulose and non-cellulosic components (including lignin, hemicellulose, and wax) [21], requiring different treatments to obtain final CNCs. These treatments reduce material weights by removing non-cellulosic components, as indicated in Table 2. Firstly, raw corn stover was washed to remove impurities and waxy substances [22], leaving 74% of the starting material (Table 2). The following alkaline treatment and bleaching got rid of hemicellulose and lignin, and the resulting samples containing cellulose were hydrolyzed using acids to eliminate amorphous regions [21] and produce CNCs (overall yield: 5%) (Table 2). A direct comparison of CNC yields in this research to the previous literature using corn stover is unavailable due to a lack of information reported. Instead, a previous study produced cellulose nanofibrils (CNFs) by oxidation treatment from the materials (washed, alkaline treated, and bleached), whose overall yield was 27.7% [19]. This high yield gap is thought to be the different composition of CNCs (consisting of only crystalline portion) from that of CNFs (containing both crystalline and amorphous regions) [23].

Corn stover-derived CNCs and intermediate products during CNC production were analyzed for their color variations (Table 2). Overall, materials became whiter after washing, alkaline, and bleaching treatments, indicated by increased L* and decreased a* and b* values (Table 2), as in a previous study [19]. However, acid hydrolysis resulted in the reduced whiteness from the bleached samples (decreased L*, increased a*, and similar b*) even though those acid hydrolyzed samples were still whiter than prebleached samples (increased L* and decreased a* and b*) (Table 2). This reduction in whiteness in the final CNCs might be because of the partial carbonization by concentrated sulfuric acids reacting with organic corn stover at high temperatures [24].

### 3.2. Characterization of Corn Stover-Derived CNCs by Acid Hydrolysis

Appearance of CNC suspension (2%, *w/w*) was shown in Figure 1a. This suspension containing CNCs from corn stover formed a colloidal structure without phase separation or precipitation (Figure 1a), and its appearance was very similar to previously reported CNC suspensions made from bagasse [25] or cotton [26]. Repulsive intermolecular forces by negatively charged sulfate groups on the surface of CNCs during acid hydrolysis were suggested as a reason for the high stability of CNC suspension [27]. 

TEM was selected to evaluate the size of individual CNC fibers. Individual CNC fibers were a rod-like structure and well dispersed with a partial aggregation (Figure 1b). Moreover, their diameter and length were around 9.0 nm and 140 nm (Figure 1b). This needle-like morphology of CNCs derived from corn stover in this research was similar with previous studies from different corn parts, including corn cob [28], corn husk [29], and corn stalk [30]. Even though there is a high similarity in morphologies, the size of individual CNC fibers was variable among different studies. This might be due to different sources of CNCs and/or different hydrolysis conditions, such as acid concentration and reaction time [28].

Hydrodynamic size distribution and surface charge of CNC fibrils was analyzed using a dynamic light scattering technique. The suspension (2%, *w/w*) containing corn stover-derived CNCs showed a broad monomodal peak with a small second peak at the tail, indicating 177 ± 2 nm as an average size (Figure 2). A polydispersity index (PDI) of this suspension was 0.43 ± 0.01 (Figure 2), indicating the slightly wide distribution of sizes (The closer the PDI value is to 0, the more homogeneous the sample is; while values closer to 1 indicate that the sample has a wide size distribution.) [31]. Moreover, its zeta potential was −39 ± 1 mV (Figure 2), corresponding to the presence of negative charge sulfate groups on the surface of CNC particles with indication of suspension stability [26].

### 3.3. Characterization of Coating Solution

Particle size distribution, closely relevant to the final coating properties, was analyzed for PVAs, which are the main ingredient of coating solutions, and their alteration was tracked after adding CNCs and beeswax (Figure 3). Those suspensions containing only PVA showed a broad size distribution (PDI 0.35) with three different peaks (mean particle size 214 nm) (Figure 3), which might be explained by the naturally high polydispersity of PVA [32]. The addition of CNCs to PVA solutions improved particle size distribution to become more monomodal (mean particle size 152 nm) with broad distribution (PDI 0.36) (Figure 3), and this distribution was highly similar to that of CNC suspension (Figure 2). The removed portion of low-size particles in the PVA solution was thought to participate in a network structure formed within PVA and CNC particles by strong hydrogen interactions [8]. After combining the mixture with beeswax, the overall particle size distribution profile was shifted to larger sizes (mean particle size 1359 nm) with continuous broad distribution (PDI 0.42) (Figure 3). This might be due to the formation of emulsion particles by aqueous immiscible beeswax trapped in a PVA and CNC network. A similar phenomenon was reported in a previous study, immobilizing large beeswax particles in a CNC network [33].

For its relation to final coating characteristics, the shear viscosity of PVA solution with/without CNCs and beeswax was measured at room temperature in order to evaluate the effect of adding CNCs and beeswax on the viscosity of coating solutions (Figure 4). The viscosity of coating solution containing only PVAs showed the constant viscosity around 0.007 Pa·s regardless of shear rates, indicating Newtonian flow (Figure 4). Generally, a synthetic polymer such as PVAs is considered as a non-Newtonian fluid, but Newtonian flow of PVAs, which is similar to this study, was also reported depending on factors such as pH and molecular weight in a previous study [34]. The addition of CNCs increased the viscosity of the coating solution containing PVAs, and the utilization of additional beeswax slightly increased the viscosity again (Figure 4). This enhanced viscosity by adding CNCs was explained by a network structure formed by strong hydrogen interactions between PVAs and CNCs [8]. In the case of the slight increase in viscosity by beeswax is thought to be due to added solid materials. Similarly, a previous study reported the viscosity increase by beeswax in starch solution due to the introduction of a new hydrophobic phase to a continuous phase [15].

### 3.4. Characterization of Coating Films

After casting coating solutions, the dried films were visually observed, and their images were displayed (Appendix A). Both coating films consisting of only PVAs and PVAs with various concentrations of CNCs were transparent, but PVA films were wobblier before adding CNCs (Appendix A). This might be due to the film reinforcement by network structure formed within PVA and CNCs [8], as explained above. An addition of beeswax reduced the transparency of final coating films even though they still maintained their slight transparency (Appendix A). This similar reduction in transparency by beeswax was reported in a previous study [15]. Those PVA films containing CNCs and beeswax showed a minor droplet aggregation and coalescence but still showed relatively homogeneous films without commercial emulsifiers (Appendix A). This might be due to the Pickering emulsion stabilizing properties of added CNCs [35]. The alterations of the amount of CNCs did not result in noticeable differences in the coating films containing PVA and beeswax based on visual inspection (Appendix A).

The thicknesses of the coating films were measured for different formulations (Table 3). In general, higher solid content was related to the increase in thickness, but overall films showed similar thickness, ranging from 0.10 to 0.16 mm (Table 3). As a result, the final film thickness can be more adjustable by the modification of other factors, such as a coating solution volume, according to the targeted applications.

The morphologies of different PVA coating films, which were visually inspected, were analyzed in more detail by an optical microscope to see the effects of CNCs and beeswax (Figure 5). The coating film containing PVAs only exhibited a homogeneous and smooth surface (Figure 5a), which was similar to previously reported results [36]. The addition of CNCs into PVA coating films showed a good distribution of CNC particles in the PVA matrix without agglomeration (Figure 5b), supporting the potential of CNCs as nanofillers to reinforce coating films. The addition of beeswax into these coating formulations produced lots of small beeswax particles, which were well dispersed in the PVA-CNC matrix (Figure 5c). These morphologies corresponded to the results of visual inspection (Appendix A), supporting CNCs as an emulsion stabilizer. 

Transmittance spectra in the UV and visible regions of PVA coating films with or without CNCs and beeswax were obtained to evaluate their transparency and shown in Figure 6. The PVA-only coating film showed the highest transmittance and its maximum transmittance for this film was up to 90% (Figure 6). The addition of CNCs to PVA coating films maintained high transparency (up to 87%) in visible ranges, even though slight reduction in transmittance in UV ranges (before 400 nm) (Figure 6). Beeswax reduced transmittance of coating films in both UV and visible ranges more than CNCs (Figure 6). However, these films containing beeswax still maintained high levels of transparency, indicating up to 74% maximum transmittance (Figure 6). These results corresponded to the previous visual inspection results (Appendix A). 

Because mechanical properties determine the coating’s ability to withstand strain imposed, the effects of CNCs and beeswax on the mechanical properties of PVA coating films were analyzed using tensile strength and Young’s modulus (Figure 7). The addition of CNCs significantly increased both tensile strength and Young’s modulus of PVA coating films (*p* < 0.05) (Figure 7). This indicates reinforcing and toughening effects on coating films by CNCs, suggesting the possible interaction between PVAs and CNCs. This result corresponds to our results above, and a similar pattern was also observed from a previous study [37]. However, loading beeswax showed different mechanical properties, maintaining a similar level of tensile strength but higher Young’s modulus compared to PVA only films (Figure 7). This increased toughness by enhanced Young’s modulus might indicate the existence of a network between PVAs and CNCs. Instead, the disturbance of beeswax on a PVA-CNC network formation is suggested to be the reason for the reduced tensile strength, which corresponded to the homogenous beeswax emulsion droplets within the coating films (Figure 5). Because high tensile strength is related to enhanced strength and durability of films, the addition of beeswax to PVA/CNC films might require a higher amount of coating mass to increase film thickness and compensate for its reduced tensile strength, depending on applications. 

WVTR is one of the most important factors in food packaging. The WVTR of PVA films with/without CNCs and beeswax was analyzed to see if they could properly enhance barrier properties of PVA films against water vapor (Figure 8). Coating films containing PVAs only showed the highest WVTR value (25 g h^−1^ m^−2^) (Figure 8). The addition of CNCs to this PVA coating film reduced the WVTR value (20 g h^−1^ m^−2^) even though its reduction was insignificant (*p* > 0.05) (Figure 8), indicating the enhancement of the barrier property of coating films. This enhanced barrier property by CNCs can be related to the result of the strong network between PVAs and CNCs, as explained above. The addition of beeswax significantly improved the barrier property of coating films (WVTR of 9 g h^−1^ m^−2^) (*p* < 0.05) (Figure 8). Beeswax has been known to be an agent that decreases water transport and increases the moisture resistance of coatings [38]. From the results, the enhancement of the barrier property of PVA coating films was obtained by adding CNCs and beeswax, indicating the improved potential of the final films for food packaging applications.

Because water vapor permeability is related to water solubility and diffusion of the water molecules through the film matrix [39], different methods that can modify these film properties, such as film density and crystalline orientation [40], can be utilized to control WVTR. In this context, the physical enhancement of PVA films by CNCs could lead to the improvement of the water vapor barrier properties of films. Based on similar highly hydrophilic properties of PVAs and CNCs, it was expected that their adequate interactions would lead to the enhancement of the final film properties in terms of mechanical and water vapor barrier properties. However, unlike hydrophobic films, water molecules could still interact with those components in PVA/CNC films and plasticize the PVA/CNC matrix, resulting in the enhanced segmental movements of polymer chains and vapor diffusion [41]. This phenomenon necessitates the inclusion of hydrophobic molecules in the final film composition. In this case, using beeswax, the additional water vapor barrier effect on PVA/CNC films was uncertain because its addition might not only impart the hydrophobicity of PVA/CNC films but also reduce intermolecular interaction and negatively affect film physical stability. However, the results showed that beeswax blending with PVA/CNC films highly improved water vapor barrier properties while maintaining the overall mechanical strength of PVA films to some degree. This phenomenon might be due to the cross-linking between the hydrophilic film components and hydrophobic groups such as C=C and C=O [42]. 

To our best knowledge, this research is the first to blend beeswax with PVAs and CNCs to enhance the final hydrophobicity and barrier effect in a moist environment. A similar previous study showed the improved water vapor barrier property of films by blending natural polymer chitosan with cellulose nanofibrils and beeswax [13]. Because of the different composition and ratio of coating formulations, it might be hard to compare our results directly. However, the coating films in the current research can have distinct advantages, such as low preparation cost and high strength resulting from using synthetic polymers instead of natural polymers.

## 4. Conclusions

Corn stover-derived CNCs as prepared by sulfuric acid hydrolysis showed needle-like morphologies (averaged diameter and length: 9.0 and 140 nm, respectively) and negative surface charges (zeta potential: −39 mV). The addition of corn stover-derived CNC and beeswax to PVA caused an increase in the particle size and viscosity of the coating solution. The addition of CNCs and beeswax to PVA showed high compatibility as the film had high homogeneity through visual and optical microscopic inspections. The light transmittance of PVA films in UV and visible ranges was gradually reduced by the addition of CNCs and beeswax. However, the PVA/CNC/beeswax films could still maintain up to 74% transmittance under the visible light, indicating the high transparency of the films. The addition of CNCs increased both tensile strength and Young’s modulus of PVA films, while further inclusion of beeswax in PVA/CNC films reduced tensile strength of the film to the level of PVA films. Water vapor barrier properties of PVA films were improved by the addition of CNCs, while significant enhancement was observed with the presence of beeswax. In a nutshell, the addition of CNCs could provide moderate water vapor protection with high tensile strength and Young’s Modulus to PVA films, while the addition of CNCs and beeswax resulted in significantly higher water vapor protection and enhanced toughness to PVA films though no significant improvement in tensile strength of the films was observed. Findings from this study indicate that CNC and beeswax could be a great addition to PVA films to enhance their properties in a moist environment, which is useful for applications in fields such as food packaging. In addition, the improved properties of these formulations suggest the potential to blend hydrophilic and hydrophobic materials together to obtain desired final properties of films and widening their applications.

## Figures and Tables

**Figure 1 polymers-15-04321-f001:**
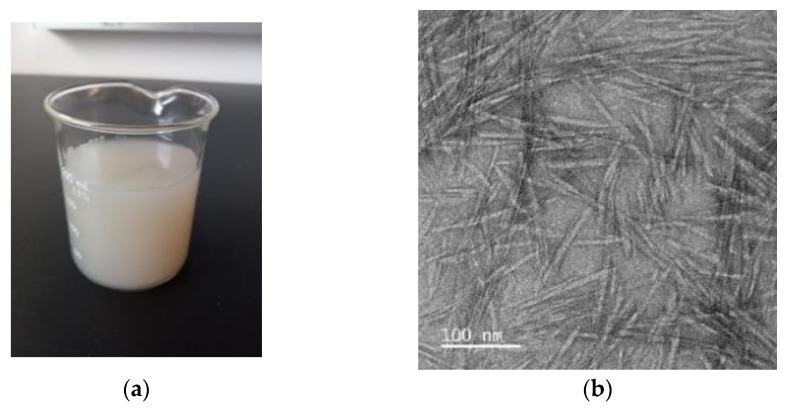
Macroscopic and microscopic images of cellulose nanocrystals (CNCs). (**a**) CNCs suspension (2%, *w/w*) in water; (**b**) A transmission electron microscopic image of CNCs.

**Figure 2 polymers-15-04321-f002:**
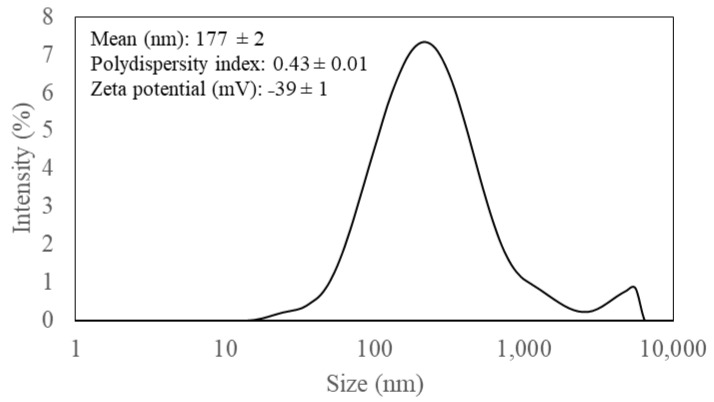
Particle size distribution of corn stover derived cellulose nanocrystals suspension (2%, *w/w*).

**Figure 3 polymers-15-04321-f003:**
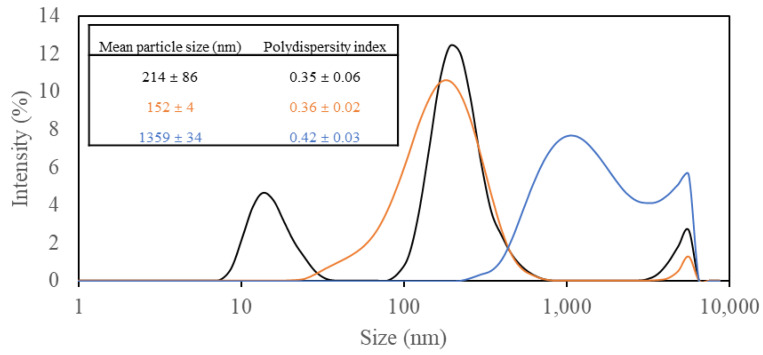
Particle size distribution of different coating solutions, including (1) 3% (*w/v*) polyvinyl alcohols (PVAs) (Black), (2) 3% (*w/v*) PVAs with 0.3% (*w/v*) cellulose nanocrystals (CNCs) (Orange), and (3) 3% (*w/v*) PVAs with 0.3% (*w/v*) CNCs and 1.5% (*w/v*) beeswax (Blue).

**Figure 4 polymers-15-04321-f004:**
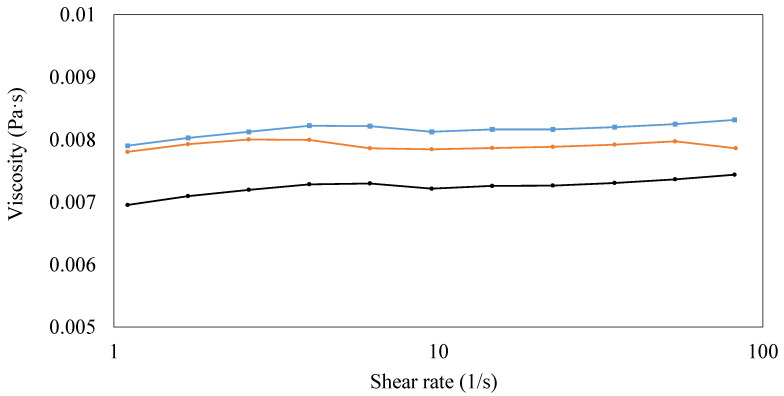
Viscosity of coating solutions. (1) 3% (*w/v*) polyvinyl alcohols (PVAs) (Black), (2) 3% (*w/v*) PVAs with 0.3% (*w/v*) cellulose nanocrystals (CNCs) (Orange), and (3) 3% (*w/v*) PVAs with 0.3% (*w/v*) CNCs and 1.5% (*w/v*) beeswax (Blue).

**Figure 5 polymers-15-04321-f005:**
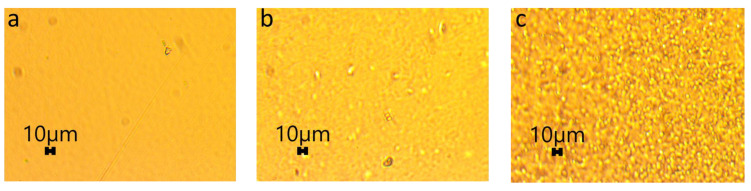
Morphologies of different coating films. (**a**) 3% (*w/v*) polyvinyl alcohols (PVAs), (**b**) 3% (*w/v*) PVAs with 0.3% (*w/v*) cellulose nanocrystals (CNCs), and (**c**) 3% (*w/v*) PVAs with 0.3% (*w/v*) CNCs and 1.5% (*w/v*) beeswax.

**Figure 6 polymers-15-04321-f006:**
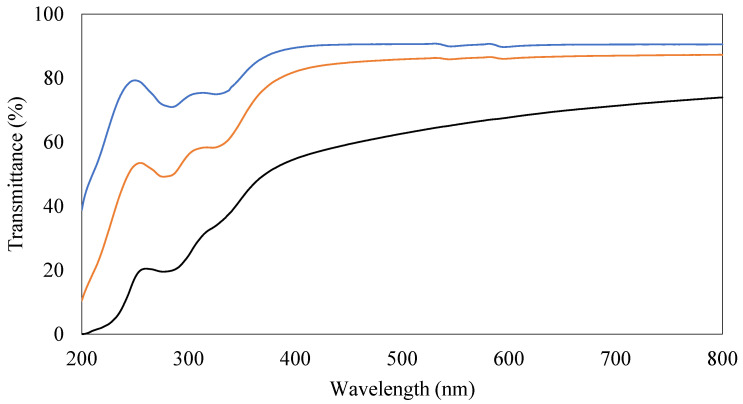
Transmittance spectra of different coating films. (1) 3% (*w/v*) polyvinyl alcohols (PVAs) (Black), (2) 3% (*w/v*) PVAs with 0.3% (*w/v*) cellulose nanocrystals (CNCs) (Orange), and (3) 3% (*w/v*) PVAs with 0.3% (*w/v*) CNCs and 1.5% (*w/v*) beeswax (Blue).

**Figure 7 polymers-15-04321-f007:**
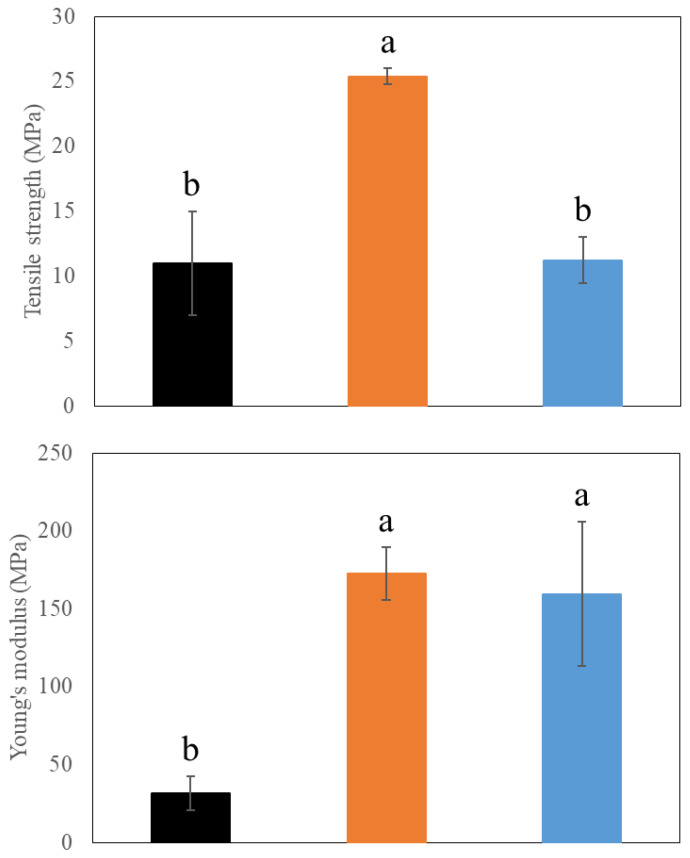
Mechanical properties of different coating films. (1) 3% (*w/v*) polyvinyl alcohols (PVAs) (Black), (2) 3% (*w/v*) PVAs with 0.3% (*w/v*) cellulose nanocrystals (CNCs) (Orange), and (3) 3% (*w/v*) PVAs with 0.3% (*w/v*) CNCs and 1.5% (*w/v*) beeswax (Blue). Different letters indicate a statistically significant difference (*p* < 0.05).

**Figure 8 polymers-15-04321-f008:**
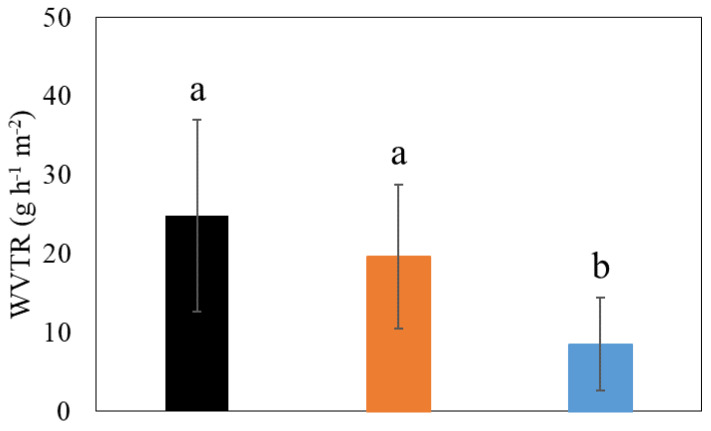
The water vapor transmission rate (WVTR) of different coating films. (1) 3% (*w/v*) polyvinyl alcohols (PVAs) (Black), (2) 3% (*w/v*) PVAs with 0.3% (*w/v*) cellulose nanocrystals (CNCs) (Orange), and (3) 3% (*w/v*) PVAs with 0.3% (*w/v*) CNCs and 1.5% (*w/v*) beeswax (Blue). Different letters indicate a statistically significant difference (*p* < 0.05).

**Table 1 polymers-15-04321-t001:** The composition of different coating solutions.

	PVA * (g)	Glycerol (g)	Beeswax (g)	CNC (2%, *w*/*w*) (g)	DI Water (mL)	Final CNC Ratio (%, *w*/*v*)
1	3	0.6	0	0	100	0
2				3.8	96.2	0.075
3				7.5	92.5	0.15
4				15	85	0.3
5			1.5	3.8	96.2	0.075
6				7.5	92.5	0.15
7				15	85	0.3

* PVA indicates polyvinyl alcohol.

**Table 2 polymers-15-04321-t002:** Yields and color of corn stover sample over different treatments for the production of cellulose nanocrystals.

	Raw	Washing	Alkaline Treatment	Bleaching	Acid Hydrolysis
Step yield (%)	N/A	74 ± 3	48 ± 1	69 ± 10	20 ± 5
Overall yield (%)	N/A	74 ± 3	35 ± 2	24 ± 4	5 ± 1
L*	30.4 ± 2.6 ^b^	32.7 ± 2.5 ^b^	31.8 ± 5.0 ^b^	43.2 ± 1.5 ^a^	34.4 ± 2.5 ^ab^
a*	3.6 ± 0.8 ^a^	3.9 ± 0.9 ^a^	2.9 ± 1.4 ^ab^	0.2 ± 0.1 ^c^	1.3 ± 0.0 ^b^
b*	13.1 ± 0.6 ^a^	12.9 ± 0.8 ^a^	11.6 ± 2.4 ^a^	3.1 ± 0.2 ^c^	5.3 ± 0.3 ^b^

Different letters indicate a statistically significant difference within each row (*p* < 0.05).

**Table 3 polymers-15-04321-t003:** Thickness of the different coating films.

Samples	Composition			Film Thickness (mm)
	Polyvinyl Alcohol (%, *v/v*)	Cellulose Nanocrystal (%, *w/v*)	Beeswax (%, *w/v*)	
1	3	0	0	0.10 ± 0.01
2		0.075		0.12 ± 0.01
3		0.15		0.13 ± 0.02
4		0.3		0.13 ± 0.02
5		0.075	1.5	0.13 ± 0.03
6		0.15		0.14 ± 0.02
7		0.3		0.16 ± 0.02

## Data Availability

Data will be made available on request.

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
