# Peer review of "Enhancing the Properties of Polyvinyl Alcohol Films by Blending with Corn Stover-Derived Cellulose Nanocrystals and Beeswax"

_polymers, 2023, doi:10.3390/polym15214321_

Round 1

Reviewer 1 Report

Comments and Suggestions for Authors

In this paper, the incorporation of beeswax into PVA/CNC films is studied, and the effects of CNC and beeswax on the properties of coatings and films are studied. The work is simple and routine, BUT gives some interesting results. you should add more in-depth mechanism introductions and explanations to highlight your scientific expertise for further consideration after minor revision.

1. Adding beeswax to PVA/CNC film reduces the tensile strength of the film, and the specific properties required for the film are not clearly described.

2. Your article's images, table layout, etc. need to be more specialized

3. You should compare this experiment with similar articles and improve it to highlight its advanced nature.

Comments on the Quality of English Language

see comments

Reviewer 2 Report

Comments and Suggestions for Authors

This manuscript presented an interesting study about the enhancing the properties of polyvinyl alcohol films by blending with corn stover-derived cellulose nanocrystals and beeswax. In my opinion, the work has potential. However, some minor points listed below need to be improved.

Abstract: please add numerical results to the abstract.

Table S1: I suggest that this table be merged into the main manuscript. I think that is not necessary presented Table S1 as a supplemental material. In addition, better comment/discuss the results presented in Table S1 in the manuscript.

Figure 5: I suggest improve Figure 5. It is hard to see any differences between the samples in this figure. Please try to make new images to better show the obtained films. I also suggest add a scale bar in the images.

A color measurement of the samples presented in Figure 5 can show differences between the samples.

Figure 6: please add a scale bar in Figure 6. Maybe the authors can add combine the images of the samples presented in Figure 5 with the morphology seen in Figure 6.

Reviewer 3 Report

Comments and Suggestions for Authors

Dear Authors,

Generally, in my opinion, this paper is interesting, and you certainly put a lot of work into it. Despite this, there are minor corrections in the text.

Detailed comments below:

Line 52: Are you sure CNC is cheap? In my opinion, this is a rather expensive addition. I suggest not writing this.

Line 70: I think the purpose of the work does not reflect the description above. You write about poor hydrophobic properties, and the purpose of your work refers to biodegradability? The purpose of the work should be reworded.

Moreover, if boideradability studies are important, they should be better described in the introduction and reference to the scientific literature.

Line 90: Add temperature deviation (+/-). Please review the methodology in this regard.

Line 150: Add caliper accuracy, e.g. 0.05?

Line 153: What type of microscope was it? (e.g. biological). You can also specify what approximation you used.

Figure 6: You should include a scale in the photos. (e.g. a short section of a given length).

Line 166: You should provide the testing standard you used.

Line 183: I understand that the test results obtained were normally distributed?

Line 378: Whether the results obtained are close or far from the expected values. Generally, this question concerns strength and water vapor permeability tests.

Line 404: One more conclusion can be added, presenting further prospects of the obtained material.
